# Impact of Vessel Transit on Vocalizations of the Taiwanese Humpback Dolphin

Wei-Chun Hu [1], Shashidhar Siddagangaiah [1,*], Chi-Fang Chen [1] and Nadia Pieretti [2]

1    Underwater Acoustic Laboratory, Department of Engineering Science and Ocean Engineering, National Taiwan University, Taipei 10617, Taiwan; william_hu@outlook.com (W.-C.H.); chifang@ntu.edu.tw (C.-F.C.)
2    Department of Life and Environmental Sciences, Polytechnic University of Marche, 60131 Ancona, Italy; nadia.pieretti@gmail.com
*    Correspondence: shashi.18j@gmail.com

**Abstract:** Recent offshore windfarm development has led to increased vessel traffic in the Eastern Taiwan Strait, which is part of the habitat of the critically endangered Taiwanese humpback dolphin (*Sousa chinensis taiwanensis*). However, data on possible effects on the behavior of this endemic subspecies are lacking to date. In this study, we observed Taiwanese humpback dolphins' acoustic behavior associated with shipping noise and analyzed their whistles and clicks before, during, and after vessel transit. Before vessel transit, the median rate of dolphin whistles and clicks was 100 and 1550 counts per minute, which significantly reduced to less than 8 and 170 counts per minute during and after vessel transit. Dolphins produced significantly shorter whistles during (0.07 s) and after (0.15 s) vessel transit. The vocalizing behavior of the Taiwanese humpback dolphin may be affected by vessel transit, which, if sustained, could possibly influence the individual communication and feeding success of the population. Implementing measures such as re-routing of the vessel lanes and regulating the speed of the vessel traffic in the habitat are proposed to overcome the influence of vessel noise on Taiwanese humpback dolphins.

**Keywords:** shipping noise; marine mammal; acoustic masking; behavioral response; Taiwanese humpback dolphin distribution





## 1. Introduction

In the last decade, the coastline of the Eastern Taiwan Strait has witnessed extensive industrial and agricultural development [1–3]. To meet increasing energy demands and the drive toward cleaner energy, the Taiwanese government implemented a large-scale offshore windfarm project in 2017 [4]. The increase in offshore structures will inevitably pose enormous stress on marine communities [3], for instance, by intensifying vessel traffic and its associated noise [5]. Vessel noise may result in behavioral disturbances of several species [6–9].

The Taiwanese humpback dolphin, *Sousa chinensis taiwanensis*, is classified Critically Endangered in the IUCN Red List of Threatened Species [10]. This endemic subspecies inhabits the coastal water of the Eastern Taiwan Strait, usually at a depth lower than 20 m [11]. The population has shown a continual decrease [10,11]. There is an urgent need to protect this species and prevent or mitigate possible threats that might impede the long-term viability of the Taiwanese humpback dolphin population.

Dolphins rely heavily on acoustics for communication, navigation, socializing, defense, predation, foraging, and reproduction [12–14]. Taiwanese humpback dolphins are highly social and are commonly found in groups. Their social behavior is essential for successful foraging and reproduction [1,11] and, therefore, population survival. Each dolphin may identify itself with a signature signal used for individual recognition, which can include both whistles and echolocation clicks [15,16]. The Taiwanese humpback dolphins may also use these signature whistles for social interaction and linkage,

signaling position and physiological state, and rearing offspring. Taiwanese humpback dolphins also produce broadband echolocation clicks for navigation and prey and object identification [17,18].

Increasing anthropogenic activities along the Eastern Taiwan Strait may affect the Taiwanese humpback dolphin by hindering auditory senses and, consequently, interfere with social networks and disrupt foraging and reproductive success [5,18]. Studies on other dolphin species indicate that vessel noise is associated with changes in vocalization and dives, avoidance of affected areas, and masking of biological signals, leading to decreased foraging success and reduced capability to attract mates, detect threats from predators, navigate, and orientation [8,19,20]. However, the influences of transiting vessel noise on vocalizations of the Taiwanese humpback dolphin in its natural habitat are yet to be explored.

In this study, we describe the acoustic behavior of the Taiwanese humpback dolphin in the Miaoli area, an offshore windfarm with significant vessel transits, where the Taiwanese humpback dolphin has frequently been spotted [10]. We observed variations in the vocalization behavior of the Taiwanese humpback dolphin during the vessel transit. This study aims to document the different types of whistles made by the Taiwanese humpback dolphin and explore variations in whistles (i.e., types and duration) and clicks before, during, and after vessel transit.

## 2. Methods

### 2.1. Study Area and Data Collection

The study area lies in the Eastern Taiwan Strait, which experiences tropical storms, monsoons in summer and winter, and an average wind speed of 12 m/s [21–23]. The ocean floor around the area is characterized by hard sedimentary rock covered with a thin layer of sandy substrate. This area is also a part of the Formosa 1 Offshore Wind Farm project, consisting of 30 turbines to produce 120 MW of power [4]. The passive acoustic monitoring (PAM) device was installed at ~10 m from demonstration wind turbine #28, which was one of the foremost turbines constructed in the Formosa 1 Offshore Wind Farm [4].

Acoustic recordings were collected off the Miaoli coast ($24°41'27''$ N, $120°48'24''$ E) by deploying an acoustic recorder with a hydrophone (Song Meter SM4M, Wildlife Acoustics), frequency range 2–48,000 Hz moored at a depth of 18–20 m. The hydrophone was set to record continuously, with a sampling frequency of 96 kHz and sensitivity of −165 dB re: 1 V/μPa. Files were recorded in.WAV format, each file with a duration of 60 min. Data were collected from the hydrophone from 1 May to 31 July 2017, amounting to 2208 h of PAM data. We analyzed the acoustic file recorded on 10 May between 0900 and 1100 h when dolphin vocalization activity was encountered during the transit of a vessel.

### 2.2. Data Analysis

The identification of whistles and clicks was based on visual and aural analysis. Bioacoustics studies use visual characterization for identifying call types [24,25]. Recordings were visually and aurally scanned for whistles and clicks in the spectrogram display of Sonic Visualiser 4.2 [26] (FFT size = 1024; Hanning window = 50% overlap). The whistles and clicks of the Taiwanese humpback dolphin vary in the frequency range of 3–8 kHz and 10–48 kHz [27]. Clicks and whistles were counted only if their intensity on the spectrogram was 10 dB re 1 $\mu Pa^2\ Hz^{-1}$ higher than the ambient noise [28,29]. The number of whistles and clicks, with the timestamp of their occurrences in the two-hour recording, was recorded. The two-hour spectrogram was computed using PAMGuide [30] with an FFT size of 1024 points and a 1 s time segment averaged to 20 s resolution. Sound pressure levels were computed in the frequency band of 10–3500 Hz and were programmed to provide a single value for every minute.

Whistle shapes are classified based on the contours of the whistles [31,32]. In this study, the identified whistles were later classified into different types based on contour shape as follows: "flat" (whistles with constant frequency and no inflection points); "rise" (whistles with ascending frequency and no inflection points); "fall" (whistles with descending frequency and no inflection points); "U-shaped" (whistles with descendent-ascendant frequencies and one inflection point); "J-shaped" and "reverse J-shaped" (whistles with descending frequency with one inflection point and ascending frequency after inflection); and "tangent-shaped" (a tangent-shaped ascending frequency curve without inflection point). The characteristics of each whistle type were categorized based on the following parameters: duration, start, end, minimum and maximum frequency, frequency range (maximum–minimum frequency), and the number of inflection points (change of slope from negative to positive or vice versa).

A previous study [27] showed that the snaps from the shrimps overlap with the Taiwanese humpback dolphins clicking, and it was highly difficult to distinguish the clicks from snaps both in the frequency of occurrence and amplitude. Accordingly, while quantifying clicks per minute, we only counted the clicks occurring in the click train. Furthermore, we also determined click characteristics, such as the maximum, minimum, peak frequency, and frequency range.

### 2.3. Statistical Analysis

The nonparametric Kruskal–Wallis test, followed by the post hoc Bonferroni multiple comparison test was used to determine the differences in whistle types, variation in the sound pressure level, the number of clicks and whistles, and whistle duration between different stages (before, during, and after) of vessel transit. Statistical analyses were performed using the "*agricolae*" package (version 1.3-3, Felipe De Mendiburu (Lima, Peru)) in R (version 3.6.2, R Development Core Team (Vienna, Austria)) [33].

### 3. Results

#### 3.1. Dolphin Vocalization Characteristics

Visual inspection of 2208 h of recordings detected 1801 whistles. A total of 501 high-quality whistles (10 dB higher than the background noise) were selected for analysis and were classified into seven types (Type 1–Type 7) depending on the variation in time and frequency (Figure 1a–g). The shapes and characteristics of the seven whistle types are summarized (mean, standard error) in Supplementary Table S1. Whistles without inflections in contours, such as flat, rise, and fall shapes, were classified as simple types (Types 2, 3, and 5); whistles with contours with inflections and different frequency curves, such as U-shaped, J-shaped, and tangent-shaped, were classified as complex types (Types 1, 4, 6, and 7) (Figure 1; Supplementary Table S1). The click train produced by the Taiwanese humpback dolphins was in the average frequency range of 14.4 kHz (with a standard error [SE] of 1.8 kHz) to 45 kHz (SE of 2.2 kHz), with two mean peak frequencies occurring at 23.6 kHz (SE of 1.2 kHz) and 35 kHz (SE of 1 kHz) (Supplementary Figure S1); the average duration of each click in a train was 2 ms (SE of 0.1 ms).

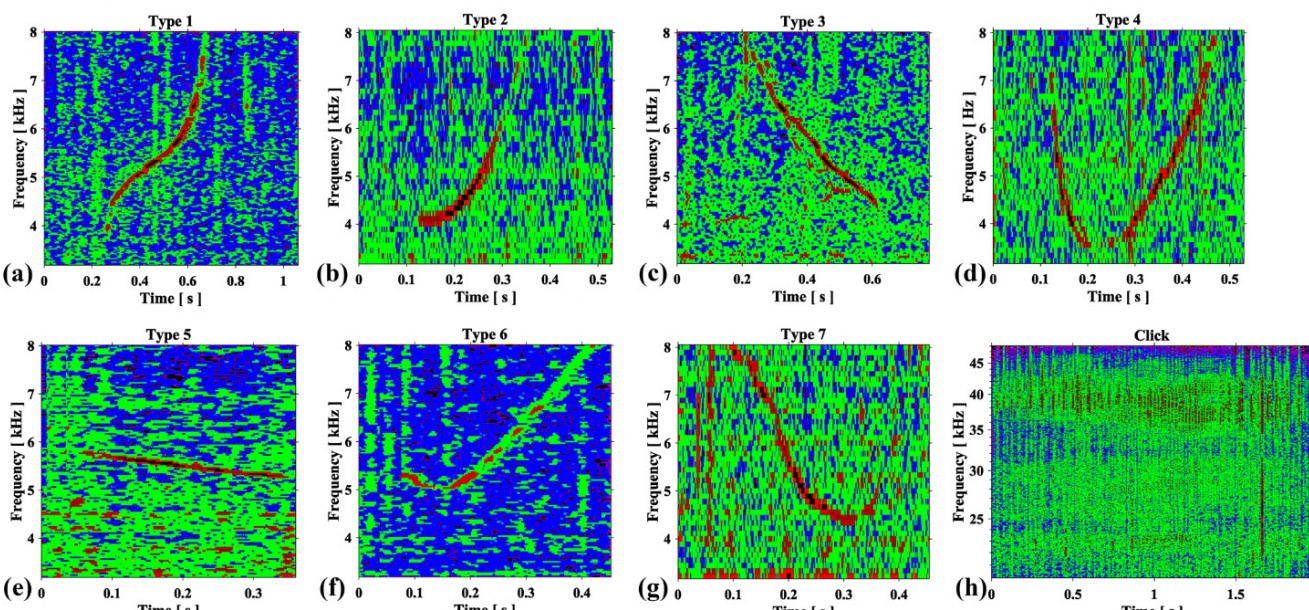

**Figure 1.** Spectrograms of whistle types (**a**–**g**) and click trains (**h**) produced by Taiwanese humpback dolphins in their natural habitat during the recording period.

### 3.2. Spectrogram Features

The visual and aural inspection of the 2208 h recordings revealed an instance of a vessel transiting amidst dolphins' vocalization on 10 May from 0900 to 1100 h. The spectrogram for the acoustic recording from 0900 to 1100 h shows noticeable high-intensity events caused by dredging at 0930 h and 1032 h (Figure 2a,b; labels D1 and D2) and the transit of two vessels (labeled on the spectrogram as V1 and V2). The spectrogram is shown in a linear scale to better visualize frequencies above 10 kHz (Figure 2b). A continual and progressively increasing click burst can be seen in the frequency band of 35–45 kHz (Figure 2b; label CB). The sound pressure level during the two-hour recording was ~130 dB re 1 μPa. However, during dredging and vessel transit (V2), sound pressure levels reached ~155 dB and ~138 dB re 1 μPa (Figure 2c,d).

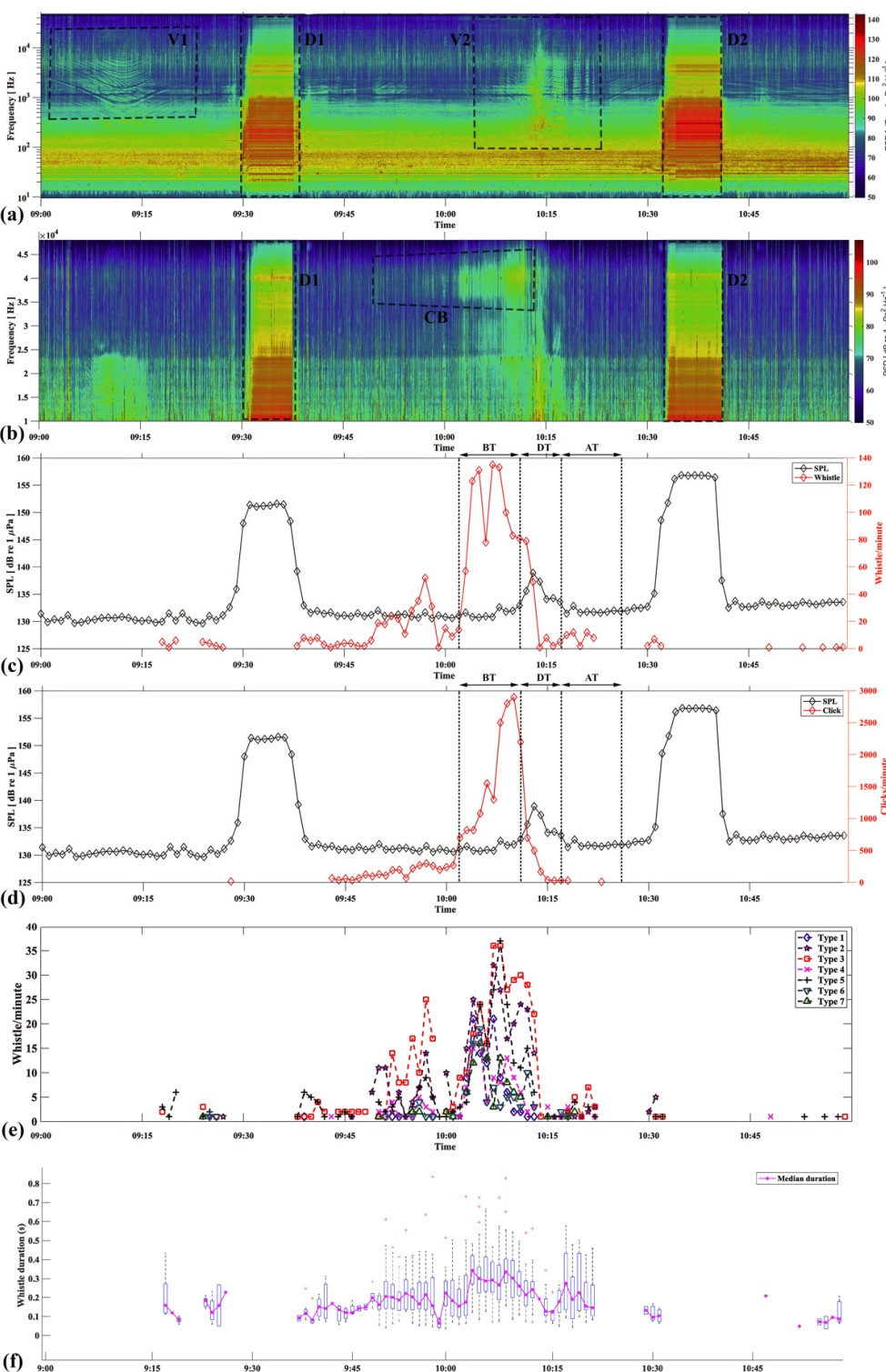

**Figure 2.** (**a**,**b**) Spectrograms in logarithmic (10 Hz–48 kHz) and linear (10 Hz–48 kHz) scale showing vessel transits (V1 and V2), dredging (D1 and D2), and click burst (CB). (**c**,**d**) Comparison of the sound pressure levels (SPL)/minute (Left axis) with the whistle/minute and the click/minute (Right axis); vessel transit is shown in the graph and is divided into before transit (BT), during transit (DT), and after transit (AT). (**e**,**f**) Variation in whistle types and duration during the two hours (Box plot: central red mark represents the median, and box limits show the 25th and 75th percentiles, whisker extremes are the maximum and minimum values, and '+' represents outliers).

### 3.3. Vocalization Activity

At 0917 h, dolphin vocalization of fewer than five whistles/minute was observed; it continued intermittently until the dredging event D1 (Figure 2c,d). No whistles were recorded during dredging, but they were observed immediately after (<20 whistles/minute and <100 clicks/minute).

### 3.3.1. Before Vessel Transit

From 1002 to 1010 h (Figure 2c,d; label BT), the whistling and clicking rates continually increased to reach a peak in whistles (135/minute) and clicks (2900/minute) at ~1007 h and ~1010 h, with Type 5 whistles occurring the most (37 per minute), followed by Types 3 and 2 (36 and 32 per minute, respectively) (Figure 2e). The median duration peaked at 0.34 s/minute (Figure 2f).

### 3.3.2. During and after Vessel Transit

During vessel transit from 1011 to 1017 h (Figure 2c,d; label DT), both whistles and clicks started to decrease (at 1017 h, <15 whistles/minute and <35 clicks/minute), with Type 3 whistles occurring the most (30 per minute), followed by Types 2 and 5 (24 and 15 per minute, respectively) (Figure 2e). The median whistle duration was 0.125 s/minute (Figure 2f). After vessel transit from 1018 to 1022 h, the whistling and clicking rate varied between 2–12/minute and 10–30/minute, respectively, with Type 3 whistles occurring the most (7 per minute), followed by Types 5 and 2 (Figure 2e). The median whistle duration was 0.27 s/minute (Figure 2f).

### 3.4. Comparison of Occurrences of Whistle Types

The occurrence of simple whistle types was significantly higher than that of complex whistle types ($p < 0.05$; Figure 3a; Supplementary Table S2). Whistle Type 3 (the median and 75th percentiles were 3 and 17 whistles/minute, respectively) was more frequently recorded, followed by Types 2 and 5. However, there was no significant difference in occurrence between complex whistle types (Types 1, 4, 6, and 7; $p > 0.05$; Figure 3a; Supplementary Table S2).

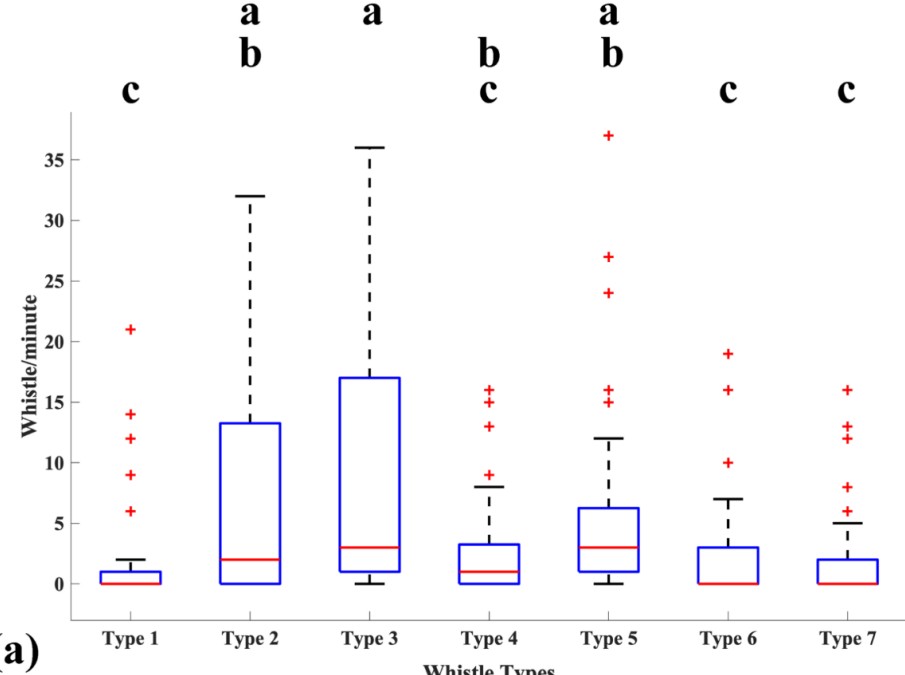

**Figure 3.** *Cont.*

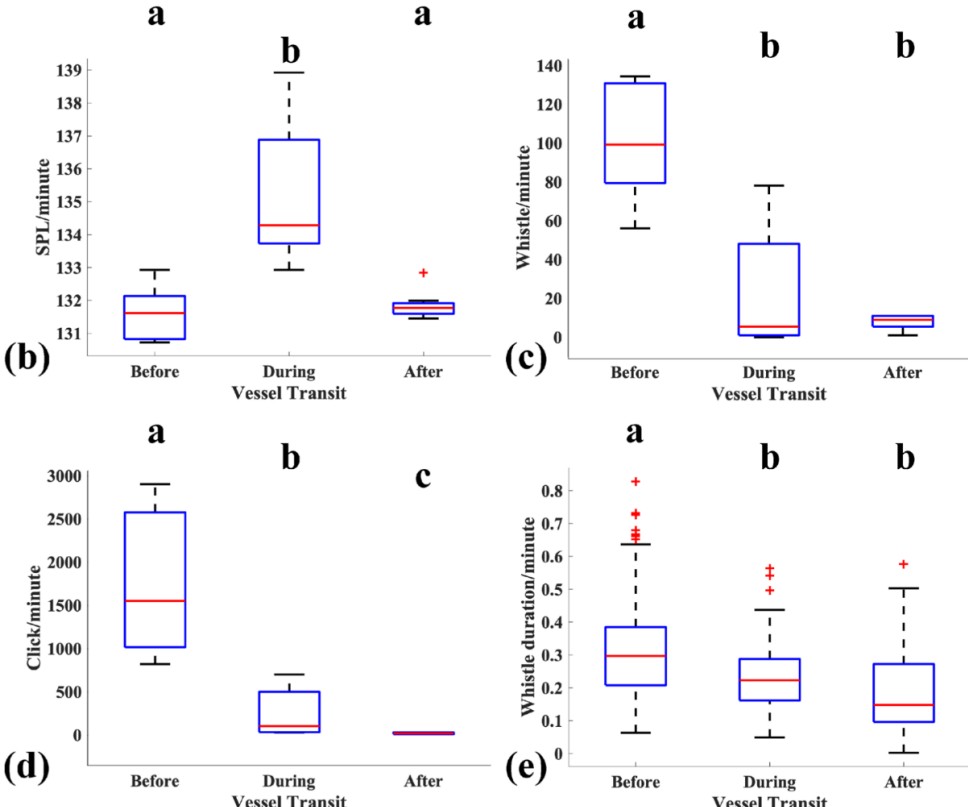

**Figure 3.** (**a**) Frequency of whistle types per minute (number of replicates in group, whistle Type 1–7; *n* = 16, 36, 45, 27, 49, 22, and 18, respectively). Effect of vessel transit (before [*n* = 9 min], during [*n* = 7 min], and after [*n* = 9 min]) on (**b**) Sound pressure level (SPL, dB re 1 μPa), (**c**) number of whistles, (**d**) number of clicks, and (**e**) whistle duration (s) within a minute. The central red mark on each box indicates the median, and the top and bottom edges of the box represent the 25th and 75th percentiles. The maximum and minimum values are marked in black at the extreme ends. The symbol '+' indicates the outliers. Superscript letters represent results of the post hoc multiple comparison test; different superscript letters indicate significant differences (*p* < 0.05), with letter 'a' at the top, and subsequent statistical differences are represented at a lower level.

*3.5. Comparison of Sound Level and Vocalizing Behavior during Various Phases of Vessel Transit*

Sound levels during vessel transit (75th percentile: 4.7 dB) were significantly higher compared to before and after vessel transit (*p* < 0.001; Figure 3b; Supplementary Table S3). The whistling and clicking rates before vessel transit (median: 100 whistles and 1550 clicks per minute) were significantly higher than those during vessel transit (median: 7 whistles and 105 clicks per minute; *p* < 0.001 and *p* < 0.01, respectively). The clicking rate significantly decreased after vessel transit (*p* < 0.05; Figure 3c,d; Supplementary Table S3). Whistle duration (median: 0.3 s per minute) was significantly higher before vessel transit than during (median: 0.22 s per minute) and after (median: 0.15 s per minute) transit (*p* < 0.001; Figure 3e; Supplementary Table S3). During all three phases of vessel transit (before, during, and after), the rate of simple whistle types (median: before = 24, during = 3.5, and after = 2 per minute) was significantly higher than that of complex whistle types (median: before = 8, during = 2, and after = 1 per minute; Supplementary Figure S2; Table S4). The occurrence of whistle Type 3 was significantly higher than that of Types 2 and 5 (*p* < 0.05); this was also observed after vessel transit (Supplementary Figure S3).

## 4. Discussion

This present study showed a significant drop in the whistling and clicking rates and significantly shorter whistles during and after vessel transit. Amid the increasing vessel traffic, shipping noise is considered a significant threat to dolphins. Changes in vocalization behavior may be due to acoustic interference, enhanced vigilance, reduced abundance, and stress. The Taiwanese humpback dolphin may respond to vessel transit by dropping its whistling and clicking rates. When noise is sustained for prolonged periods, the subsequent reduction in the calling rate may influence the dolphins' efforts to communicate and sustain social cohesion [8].

Several studies describe variation (decrease or increase) in the call rate as a response of both terrestrial and aquatic mammals to anthropogenic noise [34,35]. The Ganges river dolphin, for example, responds to vessel traffic and elevated noise levels by suppressing its acoustic activity [36]. Dolphins' call rates and click trains decreased significantly in the presence of operating vessels [37]. The bottlenose dolphins significantly produced more whistles at the onset of approaching compared to during and after vessel approaches [38]. For *Sotalia guianensis*, shipping noise from motorboats caused a significant increase in the number of whistles and a decrease in clicks [39]. Humpback dolphins significantly increased the whistle rate immediately when a boat passed through the area less than 1·5 km from the groups. In response to passing boats, groups including mother–calf pairs increased whistle rate relatively to no calves and produced significantly fewer whistles [40]. Similar differences in responses were found in bottlenose dolphin groups with or without mother–calf pairs [41]. Shipping noise was supposed to affect dolphins' communication space [8,42–44]. Dolphins may alter their vocal characteristics to avoid signal masking and to maintain communication in a noisy environment [38]. Groups including mother–calf pairs appeared to be most vulnerable to boat noise and had an increased need for communication [40]. These changes caused by shipping noise may have long-term effects [45]. The responses may sometimes be similar or different by species, group structure, behavioral state, and noise level [41,46,47]. Therefore, it would be necessary to implement both surface visual surveys and underwater acoustic monitoring for long-term data collection in the future [8,48].

The results of this study showed that Taiwanese humpback dolphins abated vocalizing behavior in response to the presence of vessel traffic. Several studies have noted the influence of vessel noise on marine mammals [8,20], such as increased metabolic stress in the Ganges river dolphin [36]. Apart from behavioral changes in marine mammals, there are instances of physical damage, including hearing loss, both temporary and permanent [49,50].

The offshore windfarm project in the Taiwan Strait began in 2016 and is planned to continue until 2030, with the capability to achieve 15 GW of power production [51]. This massive project will contribute to a substantial rise in vessel traffic and construction activities, contributing to elevated noise levels and the risk of vessel strikes in the habitat of the Taiwanese humpback dolphin. The impact of piling noise on marine mammals, especially Taiwanese humpback dolphins, has been of great concern in the environmental impact assessment of the development of offshore windfarms in Taiwan. The Taiwanese government restricts underwater noise to a sound exposure level (SEL) of no more than 160 dB (re 1 $\mu$Pa$^2$s) at a distance of 750 m from the piling. In order to ensure this standard, noise mitigation measures such as bubble curtains are taken up, which are supposed to lower the risk of a temporary hearing threshold shift (TTS) or a permanent threshold shift (PTS) in humpback dolphins. However, the behavioral effects are still unclear due to the limited studies. The whistle of the humpback dolphins is susceptible to auditory masking by piling noise, which can negatively impact the social behavior of the species [52]. The intense vessel traffic in the waters of west Hong Kong is believed to be behaviorally and acoustically disruptive to humpback dolphins [53]. The contributions in 8 and 50 kHz third-octave bands of ship noise are estimated to be auditorily sensed by and potentially affect the dolphins [54].

During the vessel transit, there is a possibility that the vessel noise in the frequency range 10 Hz−10 kHz may have masked the whistles which occur in the frequency range 3–9 kHz (Figure 1), which may affect the whistle count during the vessel transit. However, the count of the click train is not affected by the vessel noise since the click train occurs at frequencies above 20 kHz (Supplementary Figure S1). Automation algorithms developed for the detection of the dolphin whistles have also noted that the vessel noise may affect the detection accuracy [27]. Hence, to overcome this issue, in this study, we have used the visual and aural inspection of the spectrogram.

The offshore windfarm project is regarded as a threat to the Taiwanese humpback dolphin [5,10], and it could potentially induce stress on the habitats of humpback dolphins. Hence, suitable measures are required to identify the sensitive zones in the Taiwanese humpback dolphin habitat and implement speed control measures with dedicated vessel routes set up outside the habitat. Restriction of vessel speed is one of the most reliable actions to avoid physical damage to acoustic interference in marine mammals [9,55].

The vocalization response of the Taiwanese humpback dolphin to the vessel transit described in this study supports the expanding literature on the influence of vessel noise on dolphin vocalization. This study found that dolphins respond to vessel transit by decreasing their calling rate and producing shorter calls with a simpler frequency pattern. These findings also suggest that dolphins may (1) cease vocalization to avoid acoustic interference and (2) not be able to detect calls due to the increasing noise from an approaching vessel and move to other locations. However, the categorization of Taiwanese humpback dolphin call types and responses to noise is still rarely explored. In addition, acoustic responses and call types produced for social cohesion and as a reaction to stress merit further investigation. With the increasing offshore development activities in the Eastern Taiwan Strait, the ambient noise levels are likely to elevate in the future due to activities such as pile driving, construction, and vessel traffic. A deeper understanding of the Taiwanese humpback dolphins' vocalization behavior during their regular activities, such as navigation, socializing, and rearing offspring, will enable us to understand if any deviation in their vocalization behavior may be due to the impact of the anthropogenic activities. The findings of this study provide the first step to gaining an understanding of the responses of the Taiwanese humpback dolphin to vessel transit. Further studies can be carried out to understand the vocalization behavioral response of the Taiwanese humpback dolphins to the pile driving and other construction-related activities in the developmental areas at Eastern Taiwan Strait, which overlaps with the habitat of the Taiwanese humpback dolphins.

**Supplementary Materials:** The following supporting information can be downloaded at: https://www.mdpi.com/article/10.3390/d14060426/s1, Figure S1: PSD of individual click in a click train; Figure S2: Comparison of simple (Type 2, 3 and 5) and complex whistle types (Type 1, 4, 6, 7) occurring before (*n* = 27, 36), during (*n* = 12, 10) and after (*n* = 14, 6) the vessel transit. On each box, the central red mark depicts the median, and the top and bottom edges of the box represent the 25th and 75th percentiles; Figure S3: Effect of vessel transit (Before, during and after) on whistle types (1-7). On each box, the central red mark depicts the median, and the top and bottom edges of the box represent the 25th and 75th percentiles; Table S1: Mean ± SE time-frequency parameters of whistle types (*n* = 45) from Taiwanese white dolphin; Table S2: Results of Kruskal–Wallis test and the post hoc Bonferroni's multiple comparisons test showing the significance of differences in the whistle types (Type 1- Type 7) from Taiwanese white dolphin; Table S3: Results of Kruskal–Wallis test and the post hoc Bonferroni's multiple comparisons test showing the significance of differences in the SPL, whistle and clicking rate, and whistle duration occurring before, during, and after the vessel transit; Table S4: Results of Kruskal–Wallis test and the post hoc Bonferroni's multiple comparisons test showing the significance of differences between simple (Type 2, 3 and 5) and complex whistle types (Type 1, 4, 6, 7) occurring before, during, and after the vessel transit.

**Author Contributions:** W.-C.H. conducted the field work, data collection and helped S.S. in data analysis and editing the manuscript; S.S. designed the research, performed the analysis, prepared the figures and drafted the manuscript; C.-F.C. supervised the manuscript and acquired the funds; N.P. helped in the interpretation, discussion of the data, writing and editing the manuscript. All authors have read and agreed to the published version of the manuscript.

**Funding:** This research was funded by the Ministry of Science and Technology (MOST) under national energy program phase II (NEPII) grant number MOST 106-3113-E-002-011-CC2.

**Institutional Review Board Statement:** Data were conducted in the sea by deploying an autonomous passive recorder. The authors claim that this deployment did not cause any hindrance to any species or the environment. The experiments were approved by the National Taiwan University.

**Data Availability Statement:** The authors declare that the data supporting the findings of this study are available within the article and its supplementary information files, or are available from the corresponding authors upon request.

**Acknowledgments:** This research was supported by the Ministry of Science and Technology (MOST) under national energy program phase II (NEPII) grant number MOST 106-3113-E-002-011-CC2. Authors thank Rohini. B for her assistance in counting the whistles and clicks throughout the acoustic recordings.

**Conflicts of Interest:** The authors declare no conflict of interest.

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
