# Peer review of "Impact of Vessel Transit on Vocalizations of the Taiwanese Humpback Dolphin"

_diversity, doi:10.3390/d14060426_

Round 1

Reviewer 1 Report

Overall, I think and expect this is an interesting and important paper. The results are sound and implications from results can contribute to humpback dolphin conservation in the context of shipping-lane regulation around major habitats. Meanwhile, this study showed vessel movement during piling phases of OWF construction can pose potential risks to coastal delphinids nearby the turbine foundations. I sincerely recommend to accepting and publishing this paper to the journal 'Diversity' 

Author Response

We would like to take the opportunity to thank both the reviewers for the effort they put into reviewing our manuscript, which we believe is now greatly improved. All comments are reproduced below in shaded bold font, our are responses in regular font, changes in the text are in blue, and the changes in the manuscript are in shaded red font. The numbers of lines in the text are referred to the revised version, where corrections are tracked. If in some points there has been a misinterpretation of the reviewers’ recommendations, please inform us and we will correct our reply.

Reviewer #1 (Remarks to the Author):

Overall, I think and expect this is an interesting and important paper. The results are sound and implications from results can contribute to humpback dolphin conservation in the context of shipping-lane regulation around major habitats. Meanwhile, this study showed vessel movement during piling phases of OWF construction can pose potential risks to coastal delphinids nearby the turbine foundations. I sincerely recommend to accepting and publishing this paper to the journal 'Diversity' 

We authors thank the reviewer for accepting our study despite pointing out some critical flaws and providing in-depth and insightful comments and suggestions to overcome those flaws. We have made significant efforts to address these concerns in the revision. In the revised manuscript, following the reviewer's specific suggestions. We have carefully added the references related to dolphin studies and have removed the references pointing to other mammals.

what projects? which kind of projects can 'significantly increased vessel

We have mentioned 'offshore projects' (Line 9)

Here, please use 'Taiwanese humpback dolphin". There is no such a specie or 'subspecies' called "Taiwanese white dolphin". This name may be used in local reports but, as this is submitted to an INTERNATIONAL journal, using an 'international recognizable' name would be better.

We have changed it all over the manuscript as 'Taiwanese humpback dolphin'

'However' should not be used here (that means you don't agree with the point in the previous sentence).

Removed it

DO NOT use 'the first time' anytime in a scientific paper. You never know whether you are the 'first one' to do this work (indeed, you are not).

We have removed it all over the manuscript

what is a 'definite' variation.

We have rewritten this as: (Line 12-13)

..we observed a Taiwanese humpback dolphins' acoustic behavior associated with shipping noise, and analyzed its whistles and clicks before, during, and after vessel transit..

which/who is 'its'?

?? if your 'its' means the 'humpback dolphin', write it down.

Corrected it as Taiwanese humpback dolphin (Line 12)

100 and 1550 of what? (i.e. what's the unit ? do you mean 100 whistles and 1550 clicks? if it is, you need to reorganize this sentence.

We have rewritten this as: (Line 14-16)

..Before vessel transit, the median rate of dolphin whistles and clicks was 100 and 1550 counts per minute, respectively, which significantly reduced to less than 8 and 170 counts per minute, respectively, during and after vessel transit..

You mention '..could possbly influence its behavioral ecology', which is is very important and meaningful. My concern/suggestion is, however, you should not end your abstract here (in this way). your have shown significant changes. What you should further go is presenting a recommendation to solutions: What's your recommendation/suggestions to mitigate such impacts (e.g. re-routing vessel lanes or speed limitation to vessel traffic??).

We have added this as follows: (Line 19-21)

Further, implementing measures such as re-routing of the vessel lanes and regulating the speed of the vessel traffic in the habitat may be taken up to overcome the influence of vessel noise on Taiwanese humpback dolphin.

My first impression to Introduction shows two critical flaws:

  1. the English quality requires substantial revisions, particularly on the context and flow.

  1. many papers that are important in the field of humpback dolphin researches, not only the Indo-Pacific humpback dolphin but also the endemic species 'Taiwanese humpback dolphin' researches, are not referenced here. They are critically important to support your points and discussions and never 'weaken' your research 'novelty'. These papers, indeed, reinforce your descriptions that this is an important research. I have no idea why you do this but this is a very serious, fatal indeed, flaw. (One question: Do you overlook those papers to highlight your 'novelty' (i.e., I am the first on to publish the Taiwanese humpback dolphin acoustic paper)? you should be very careful to this condition. In my experience of paper reviews and being a journal editor, this is treated as an ethic violation behavior. )

We thank reviewer for these suggestions:

1) We have proofread the manuscript to improve the language and flow.

2) In the revised manuscript upon the suggestions of the reviewer, we have added the references which discusses about the humpback dolphins. We apologize for missing to refer some important studies and in the revised version we have overcome this flaw. We thank you for all the important references you suggested.

Strikethrough ‘Huang 2011

Deleted this reference

I think you need to cite following two references here:

  1. Karczmarski, L., Huang, S.-L., Wong, W.-H., Chang, W.-L., Chan, S. C. Y., Keith, M. (2017). "Distribution of a Coastal Delphinid Under the Impact of Long-Term Habitat Loss: Indo-Pacific Humpback Dolphins off Taiwan’s West Coast." Estuaries and Coasts 40: 594–603. https://doi.org/10.1007/s12237-016-0146-5
  2. Huang, S.-L., Wang, X., Wu, H., Peng, C., Jefferson, T. A. (2022). "Habitat protection planning for Indo-Pacific humpback dolphins (Sousa chinensis) in deteriorating environments: knowledge gaps and recommendations for action." Aquatic Conservation: Marine and Freshwater Ecosystems 32(1): 171-185. https://doi.org/10.1002/aqc.3740

these two papers explicitly show 'extensive industrial and agricultural development' in the eastern Taiwan Strait and analyze/discuss the influence of such activities on the humpback dolphin's distribution and habitat utilization.

meanwhile, I don't think the reference 'Huang, 2011' is adequate here as this is not published on a 'peer reviewed' scientific journal. 

Thanks for suggesting these references. We have included these references. (Line 27-28)

Highlighted ‘Taiwanese’

Corrected it to ‘Taiwanese’. (Line 29)

implements (if this project is suspended or stopped now, using 'implemented' is fine).

We have rephrased it

‘the Taiwanese government implemented a large’ (Line 29)

Strikethrough ‘rapid’

Removed it.

you need a reference here, which I suggest this paper:

Huang, S.-L., Wang, X., Wu, H., Peng, C., Jefferson, T. A. (2022). "Habitat protection planning for Indo-Pacific humpback dolphins (Sousa chinensis) in deteriorating environments: knowledge gaps and recommendations for action." Aquatic Conservation: Marine and Freshwater Ecosystems 32(1): 171-185. https://doi.org/10.1002/aqc.3740.

Thanks for this suggestion. We have referred this. (Line 31)

Highlighted ‘15 GW of capacity by 2035’

We have removed this and also have removed the reference which was not a peer-reviewed

In Abstract (the opening sentence), you should present information in this sentence.

besides, in the end of this sentence, you should cite this paper:

Wright, A. J., Araújo-Wang, C., Wang, J. Y.,  Ross, P. S., Tougaard, J., Winkler, R., Márquez, M. C., Robertson, F. C., Williams, K. F., Reeves, R. R. (2019). "How ‘Blue’ Is ‘Green’ Energy." TRENDS in Ecology & Evolution 35: 235-244. https://doi.org/10.1016/j.tree.2019.11.002

We have added this reference. (Line 32)

Use 'Taiwanese humpback dolphin'  here and following sentences. DO NOT use 'Taiwane white dolphin."

meanwhile, please do not use the abbreviation of a term, such as the 'TWD' and 'ETS', as your readers may not be and are often not researchers/students in your field and not familiar with the topic you discussed.

We have replaced it as Taiwanese humpback dolphin all over the manuscript.

Hightlighted ‘ETS’

We have replaced it as Eastern Taiwan strait.

Hightlighted  ‘According to a recent survey by the Council of Agriculture, this species totals only about 50 individuals’

Deleted this.

‘Winkler 2019’  In a scientific paper, citation of the literature that is not peer reviewed should be avoid, unless there is no citable source.

Deleted this reference.

Strikethrough ‘Joan Gonzalvo 2020’

Deleted this reference.

what is the 'further spread of its population'? ''spread' should not be used here (unless you treat the humpback dolphin as 'pests').

We have corrected this as: (Line 41)

“..might impede the further growth of THD population.”

What's the difference between this two papers? meanwhile, as far as I read, Wang et al., 2007 presents the results of line-transect surveys and population-abundance estimates but not concludes the result you mentioned here. you should cite references related with the correlation between dolphin vocalization and behavioral patterns.

here, you should rewrite this sentence. the reference you cite is not specific to 'this mammal' but for all dolphins. You should not and can not cite your reference in this way.

We have removed these references and have added new references specific to dolphins. (Line 43-44)

“Dolphins relies heavily on acoustics for communication, navigation, socializing, defense, predation, foraging, and reproduction(Au 1993, Au 2000, Au and Hastings 2008).”

who is highly social??

Corrected it as Taiwanese humpback dolphin. (Line 45)

here, again, you used a 'general' reference to support the your 'highly specific' description. in a scientific paper, you can not and should not do this. following your previous discussion, the context you are talking about is the humpback dolphin, in this situation, you should cite the reference specific to this species. if there is no reference available, you should soften your tongue, such as 'may identify'.

Cheng, Z., Wang, D., Wu, H., Huang, S.-L., Pine, M. K., Peng, C., Wang, K. (2017). "Stereotyped Whistles May Be First Evidence to Suggest the Possibility of Signature Whistles in an Injured Indo-Pacific Humpback Dolphin (Sousa chinensis)." Aquatic Mammals 43: 185-192

Thanks for this suggestion,we have rewrite this sentence and  have added this reference. (Line 47-48)

Each dolphin may identifies itself with a signature signal used for individual recognition, which can include both whistles and echolocation clicks (Tyack 2000, Cheng, Wang et al. 2017)

You need to rewrite this sentence as you have no reference here. you can not conclude this sentence.

We have rewritten this sentence as: (Line 49-51)

The Taiwanese humpback dolphins may also use these signature whistles for social interaction and linkage, signaling position and physiological state, and rearing offspring.

If the word 'It' indicates the Taiwanese humpback dolphin, you need to cite following two references:

  1. Lin, T.-H., Akamatsu, T., Chou, L.-S. (2013). "Tidal influences on the habitat use of Indo-Pacific humpback dolphins in an estuary." Marine Biology 160: 1353–1363
  2. Lin, T.-H., Akamatsu, T., Chou, L.-S. (2015). "Seasonal distribution of Indo-Pacific humpback dolphins at an estuarine habitat: influences of upstream rainfall." Estuaries and Coasts 38: 1376-1384

We have cited these references. (Line 53-54)

Do 'anthropogenic activities' and 'noise' indicate different things? If not, you should rewrite this sentence. If it is, you should cite other reference here, besides "Taylor, 2019" , e.g. Lin et al., 2013, 2015; Wright et al., 2019 (that I provided previously).

We have cited these references and have rewrite the sentence as below: (Line 57-58)

Increasing anthropogenic activities along the Eastern Taiwan Strait may affect the Taiwanese humpback dolphin by hindering its auditory senses, and may, consequently, interfere with its social network and disrupt its foraging and reproductive success(Lin, Akamatsu et al. 2015, Wright, Araújo-Wang et al. 2020)

who/which is 'its'?

We have corrected it as ‘Taiwanese humpback dolphin’. (Line 56)

I don't get the flow of this sentence. If you have concluded this and this and this in the Taiwanese humpback dolphin, why you need to talk other species here. I mean, you can talk about this, but this description should be placed before your 'question'. You can talk about this sentence, that's fine, then you ask a 'question' like 'For the Taiwanese humpback dolphin' similar this and that may happen too, which results in this and that.'

The influences of transiting vessel noise on vocalizations of the TWD in its natural habitat are yet to be explored.

We have rewritten this paragraph to ask the question after the sentences specified by the reviewer. (Line 58-64)

Studies on other dolphin species have shown that vessel noise is associated with changes in vocalization and dives, avoidance of affected areas and masking of biological signals, leading to decreased foraging success and reduced capability to attract mates, detect threats from predators, navigate, and orient (Council 2003, Halliday, Insley et al. 2017, Erbe, Marley et al. 2019). However, the influences of transiting vessel noise on vocalizations of the Taiwanese humpback dolphin in its natural habitat are yet to be explored.

acoustic behavior

Rewritten the sentence as: (Line 65-66)

In this study, we describe the acoustic behavior of the Taiwanese humpback dolphin in the Miaoli area,..

Why did you just analyze a single day? Do you mean, for the other days, the situation did not happen?

I think, in your results, you should present how many days there was the situation "when dolphin vocalization activity was encountered during the transit of a vessel". This information is important to measure how important the impact you present will be.

In the revised manuscript, at the result section we have mentioned it as below: (Line 155-157)

The visual and aural inspection of the 2208 h recordings revealed an instance of a vessel transiting amidst dolphins' vocalization on May 10th from 0900 to 1100 hours. The spectrogram for the acoustic recording from 0900 to 1100 hours is shown in Figure 2a

I have no idea why this information is presented here. It seems just local information.

Yes, it’s the description of the study area. We have renamed the Section 2.1 as “Data collection and study area” (Line 74)

I have no idea why you present this information here, which is irrelevant to the topic of this study.

These two sentences are quite dangerous. If I did not read your abstract and figures, I will reject your submission immediately without any hesitation because of these two sentences.

  1. you claim 'concrete substrate ..... provides shelter and food for these fishes'. the truth is, however, this is not supported by any peer-review article. In Europe, there are some evidences showing the turbine foundations might function as 'artificial reef' that attract fishes but also alter original fauna and ecosystem functions.
  2. following above criticism, there is no such report supporting above hypothesis. The species you mentioned are indeed mostly deep-water species that seldom (bottlenose dolphin) or never (routh-toothed dolphin, pygmy sperm whale, Blainville's beaked whale) occur in shallow waters, alive. the reference you cited (Wang, Li et al., 2015) does not support your statements nor the context you presented. When you writing a paper, you should be very cautious to this kind of 'indirect citation'.

In the revised manuscript, we have removed the whole paragraph.

How 'reliable' is reliable? When you describe something in M&M, you should be very cautious to this kind of adj.

My suggestion is do no use this kind of adj unless you have sufficient statistical details to show your 'reliability'.

We have removed the phrase ‘reliable’.

just 'use' (this is a highly matured technique)

Changed it to ‘use’

identifying call types

Added this as: (Line 95-96)

“Bioacoustics studies use visual characterization for identifying call type..”

This sentence reads awkward, please rewrite it.

We have rewrite this as: (Line 121-122)

“A previous study by (Siddagangaiah, Chen et al. 2020) has shown that the snaps from the shrimps overlap with the Taiwanese humpback dolphins clicking and..”

I prefer using 'distinguish the clicks from snaps' (yet, it's up to your decision).

Rewrite this sentence as: (Line 123)

“it was highly difficult to distinguish the clicks from snaps both in the..”

'Accordingly',

Changed it to ‘Accordingly’(Line 124)

I suggest you provides the details of your sample size: how many recordings did you collected? How many recordings you collected showed the feature 'dolphin vocalization was encountered during a vessel transit'?

We have added the total number of hours analyzed: (Line 138 and 155-157)

“Visual inspection of 2208 h of recordings detected 1801 whistles”

“The visual and aural inspection of the 2208 h recordings revealed an instance of a vessel transiting amidst dolphins' vocalization on May 10th from 0900 to 1100 hours. The spectrogram for the acoustic recording from 0900 to 1100 hours is shown in Figure 2a.”

what is this?? I mean 48h of recordings are collected from when to when?

Thanks for this suggestion. We have analyzed the whole monitoring duration lasting 2208 hours of data to detect 1801 whistles. We have corrected it as follows: (Line 138)

“Visual inspection of 2208 h of recordings detected 1801 whistles.”

I suggest the author reorganize the discussion in following three directions:

  1. What do my results mean? the author did some jobs here but not well organized.
  2. what will be your recommendations or suggestions to measures that mitigate the impacts of vessel transits (and also piling). (in the meantime, I think you present an interesting and important results showing during piling, dolphin's vocalization ceased. I wonder, is there any 'noise-mitigation measure' applied to piling. if there are such measures, that means those mitigation measures may simply reduce the sound pressure but still pose chronic influences on dolphin's behaviors. you can/may develop this point to a new discussion.
  3. What's the next step? you stated some next-step studies. that's fine. but you should mention why these studies are important, based on your results.

We thank reviewer for this important suggestion

1) We have added the new paragraph describing our work with relation to the dolphins and have removed the content discussing about other mammals. (Line 243-265)

2) As suggested, we have added the potential influence of the pile driving on dolphins and the noise mitigation measures required when carrying-out pile driving. (Line 278-291)

“The impact of piling noise on marine mammals, especially Taiwanese humpback dolphins, has been of great concern in the environmental impact assessment of the development of offshore wind farms in Taiwan. The Taiwan government requires the pile-driving noise to be no more than a sound exposure level (SEL) of 160 dB (re. 1μPa2?) at 750m. To ensure this standard, the noise mitigation measures such as bubble curtains are taken up, which may be able to avoid a temporary hearing threshold shift (TTS) or a permanent threshold shift (PTS) in humpback dolphins. However, the behavioral effects are still unclear due to the limited studies. The whistle of the humpback dolphins is susceptible to auditory masking by piling noise, which can negatively impact the social behavior of the species(Wang, Wu et al. 2014). The intense vessel traffic in the waters of west Hong Kong is believed to be behaviorally and acoustically disruptive to humpback dolphins(Sims, Hung et al. 2012). The contributions in 8 and 50 kHz third-octave bands of ship noise are estimated to be auditorily sensed by and potentially affect the dolphins(Liu, Dong et al. 2017).”

3) As suggested by the reviewer we have discussed regarding our results and have suggested the further steps and importance of understanding the vocalization behavior of the dolphins. (Line 314-329)

“..However, the categorization of Taiwanese humpback dolphin call types and response to noise are still unexplored areas that need deeper study. In addition, acoustic responses and call types produced during activities (such as foraging and mating), for social cohesion, and as a reaction to stress require further investigation. With the increasing offshore developmental projects in the Eastern Taiwan Strait, the ambient noise levels are likely to elevate in the future due to activities such as pile driving, construction, and vessel traffic. A deeper understanding of the Taiwanese humpback dolphins' vocalization behavior during their regular activities, such as navigation, socializing, and rearing offspring, will enable us to understand if any deviation in their vocalization behavior may be due to the impact of the anthropogenic activities. The findings of this study provide the first step to gain an understanding of the responses of the Taiwanese humpback dolphin to vessel transit. Further studies can be carried-out to understand the vocalization behavioral response of the Taiwanese humpback dolphins to the pile driving and other construction-related activities in the developmental areas at Eastern Taiwan Strait, which overlaps with the habitat of the Taiwanese humpback dolphins.”

You can rewrite these two sentence into one.

We have rewritten this sentence. (Line 229-232)

The influence of vessel transit on the vocalization of the Taiwanese humpback dolphin in its natural habitat was investigated, and it was found that there was a significant drop in the whistling and clicking rates and significantly shorter whistles during and after vessel transit

DO NOT explain or hypothesis what you did not know without any evidence/reference support your ideas.

We have removed it.

The reference you cited is now suitable here. you are talking about dolphins, which use 'high-frequency' vocalization; the reference you cited talks about the humpback whale, which is a completely different story.

We have removed it.

Indeed, I have no idea what you are talking about in this entire paragraph. what's your main point? what's your 'take-home' conclusion. the ending of this paragraph seems not relevant to your results. when you make a discussion, it should always start and center around your results. you can go as far as you can but it still stands on your results. The most important thing is, the conclusion (ending) or 'take-home' message should always stick on your results.

again, I suggest you cite the works that talk about DOLPHINS. Bowhead whale and right whale are big baleen whales (very big one), which is a complete story. 

We have deleted the whole paragraph and have replaced with the studies dealing with only dolphins. (Line 243-265)

“Dolphins' call rates and click trains decreased significantly in the presence of operating vessels(Luís, Couchinho et al. 2014). The bottlenose dolphins significantly produced more whistles at the onset of approaching compared to during and after vessel approaches(Buckstaff 2004). For Sotalia guianensis, shipping noise from motorboats caused a significant increase in the number of whistles and a decrease in clicks(Martins, Rossi-Santos et al. 2018). Humpback dolphins had been observed to significantly increase the whistle rate immediately when a boat passed through the area less than 1·5 km from the groups. In response to passing boats, groups including mother-calf pairs increased whistle rate relatively to no calves produced significantly fewer whistles (Van Parijs and Corkeron 2001). Similar differences in responses were found in bottlenose dolphin groups with or without mother-calf pairs(Guerra, Dawson et al. 2014). Shipping noise was supposed to affect dolphins' communication space(Jensen, Bejder et al. 2009, Merchant, Pirotta et al. 2014, Putland, Merchant et al. 2018, Erbe, Marley et al. 2019). Dolphins may alter their vocal characteristics to avoid signal masking and to maintain communication in a noisy environment(Buckstaff 2004). Groups including mother-calf pairs appeared to be most vulnerable to boat noise and had an increased need for communication(Van Parijs and Corkeron 2001). These changes caused by shipping noise may have long-term effects(Heiler, Elwen et al. 2016). The responses may sometimes be similar or different by species, group structure, behavioral state, and noise level(May-Collado and Wartzok 2008, Guerra, Dawson et al. 2014, Gospić and Picciulin 2016). Therefore, it would be necessary to implement both surface visual survey and underwater acoustic monitoring for long-term data collection in the future (Marley, Salgado Kent et al. 2017, Erbe, Marley et al. 2019).”

DO NOT USE un-PEER REVIEWED PAPERS.

We have removed those referred sentences and references.

be careful to this writing, I suggest you rewrite this sentence here. Animals do not 'EXHIBIT' any behavior.

We have removed ‘exhibit’ and rewrite the sentence

The results in this study showed that Taiwanese humpback dolphins abated vocalizing behavior in response to the presence of vessel traffic.

This is fish (very delicious fish), not a marine mammal.

We have deleted it.

This paper does not talk about this.

We have removed that reference.

This paper talks about vessel collisions, not the context you are talking about. If you want to preserve this paper, you need to explicitly describe its connection. 

We have removed this reference.

; Wright et al., 2019

Added (Line 300-301)

“The offshore windfarm project is regarded as a threat to the Taiwanese humpback dolphin (Wang 2018, Wright, Araújo-Wang et al. 2020), and..”

and what is your suggestion to the context of Taiwan you are talking about. Your suggestion should also be presented in the Abstract.

We have added suggestion as in abstract (Line 19-21) and discussion (Line 303-307)

“Further, implementing measures such as re-routing of the vessel lanes and regulating the speed of the vessel traffic in the habitat may be taken up to overcome the influence of vessel noise on Taiwanese humpback dolphin.”

“..and implement speed control measures with dedicated vessel routes set up outside the habitat. Restriction of vessel speed is one of the most reliable actions to avoid physical damage to and acoustic interference in marine mammals (Rockwood, Adams et al. 2020, Schoeman, Patterson-Abrolat et al. 2020)”

what do you mean??

Rewritten the sentence. (Line 308-310)

“The vocalization response of the Taiwanese humpback dolphin to the vessel transit described in this study supports the expanding literature on the influence of vessel noise on dolphin vocalization”

you did not suggest this point in previous discussion, at least I didn't read it. You can not make your conclusion based on something you never discussed (from your results).

We have removed it.

Why this is important??

We have expanded our explanation to describe the importance of the understanding the vocalization behavior. (Line 318-329)

. With the increasing offshore developmental projects in the Eastern Taiwan Strait, the ambient noise levels are likely to elevate in the future due to activities such as pile driving, construction, and vessel traffic. A deeper understanding of the Taiwanese humpback dolphins' vocalization behavior during their regular activities, such as navigation, socializing, and rearing offspring, will enable us to understand if any deviation in their vocalization behavior may be due to the impact of the anthropogenic activities. The findings of this study provide the first step to gain an understanding of the responses of the Taiwanese humpback dolphin to vessel transit. Further studies can be carried-out to understand the vocalization behavioral response of the Taiwanese humpback dolphins to the pile driving and other construction-related activities in the developmental areas at Eastern Taiwan Strait, which overlaps with the habitat of the Taiwanese humpback dolphins.

Reviewer 2 Report

This paper has some merit and provides useful information, but has some serious (though easily fixed) problems that should be addressed before it is published. See my specific comments below:

  • Lines 35-36 – This dolphin species occurs in coastal waters, but not usually in harbors. Please correct.
  • Lines 47-51 – The authors are talking about a different species of dolphin here, the bottlenose dolphin (Tursiops truncatus). It is possible that humpback dolphins also have signature whistles, but this has not been proven.
  • Methods section – What was the frequency range of the recording equipment? Were the full range of dolphin vocalizations able to be recorded?
  • Lines 91-93 – The authors are talking about the coastal, shallow water habitat of humpback dolphins here, but many of the species they list (e.g., rough-toothed dolphins, pygmy sperm whales, beaked whales) are deepwater animals that do not occur in such shallow waters.
  • Lines 175-178 – Is it possible that some dolphin sounds were masked by the loud noises in the environment? This should be discussed.
  • Lines 214-227 – Again, there is concern that some of the quieter sounds produced by the dolphins (esp. whistles) might have been drowned out by the loud noises of vessels, and other activity in the habitat. The authors need to consider this factor in the paper, and discuss it much more extensively.
  • Literature Cited – This has serious problems. The authors cited work on other species (even other classes of animals) and try to apply it to humpback dolphins. They also cited popular literature, and new reports that have not gone through peer review. This is not acceptable in a scientific paper. Finally, there are many incomplete references, and some with clear mistakes, and inconsistent formatting. This is very sloppy and extremely poor work. The lit. cited section needs to be completely redone to resolve these issues.

Author Response

Reviewer 2

This paper has some merit and provides useful information, but has some serious (though easily fixed) problems that should be addressed before it is published. See my specific comments below:

We authors thank the reviewer for providing in-depth and insightful comments and suggestions. We have made significant efforts to address these concerns in the revision. In the revised manuscript, following the reviewer's specific suggestions. We have carefully added the references related to dolphin studies and have removed the references pointing to other mammals.

Lines 35-36 – This dolphin species occurs in coastal waters, but not usually in harbors. Please correct.

We have corrected this. (Line 37)

“..is included in the IUCN Red List as Critically Endangered (Wang 2018). It can be encountered in coastal water of the Eastern Taiwan Strait, usually at a depth lower than 20 m”

Lines 47-51 – The authors are talking about a different species of dolphin here, the bottlenose dolphin (Tursiops truncatus). It is possible that humpback dolphins also have signature whistles, but this has not been proven.

We have rewritten this sentence and have included the phrase ‘may identify’. (Line 47-52)

“Each dolphin may identifies itself with a signature signal used for individual recognition, which can include both whistles and echolocation clicks (Tyack 2000, Cheng, Wang et al. 2017). The Taiwanese humpback dolphins may also use these signature whistles for social interaction and linkage, signaling position and physiological state, and rearing offspring”

Methods section – What was the frequency range of the recording equipment? Were the full range of dolphin vocalizations able to be recorded?

The sampling frequency of the hydrophone is 96kHz and it can record in the frequency range 2 – 48000 Hz. As shown in the Figure 1 the whistles fall in the frequency range 3000-9000 Hz and the click train occur in the frequency range 14-45kHz (Supplementary Figure S1). Hence the hydrophone can record the dolphin vocalizations.

Lines 91-93 – The authors are talking about the coastal, shallow water habitat of humpback dolphins here, but many of the species they list (e.g., rough-toothed dolphins, pygmy sperm whales, beaked whales) are deepwater animals that do not occur in such shallow waters.

We have removed this content and have deleted the paragraph.

Lines 175-178 – Is it possible that some dolphin sounds were masked by the loud noises in the environment? This should be discussed.

Lines 214-227 – Again, there is concern that some of the quieter sounds produced by the dolphins (esp. whistles) might have been drowned out by the loud noises of vessels, and other activity in the habitat. The authors need to consider this factor in the paper, and discuss it much more extensively.

We thank reviewer for this important suggestion. We have added this issue of possible masking in the discussion Section. (Line 292-299)

During the vessel transit, there is a possibility that the vessel noise in the frequency range 10 Hz-10kHz may have masked the whistles which occur in the frequency range 3-9kHz (Figure 1). Thus, may affect the whistle count during the vessel transit. However, the count of the click train is not affected by the vessel noise since the click train occurs at frequencies above 20 kHz (Supplementary Figure S1). Automation algorithms developed for the detection of the dolphin whistles have also noted that the vessel noise may affect the detection accuracy(Siddagangaiah, Chen et al. 2020). Hence, to overcome this issue, in this study we have used the visual and aural inspection of the spectrogram.

Literature Cited – This has serious problems. The authors cited work on other species (even other classes of animals) and try to apply it to humpback dolphins. They also cited popular literature, and new reports that have not gone through peer review. This is not acceptable in a scientific paper. Finally, there are many incomplete references, and some with clear mistakes, and inconsistent formatting. This is very sloppy and extremely poor work. The lit. cited section needs to be completely redone to resolve these issues.

We thank reviewer for this important suggestion. We have removed all the articles which are not peer-reviewed and have also removed the references of other species and have added the references specific to dolphins. We have carefully gone through the references to check formatting and have corrected.

Round 2

Reviewer 1 Report

I am quite satisfied and appreciated to the present revision. It's flow is quite smooth and reads comfortable now. Congrats. Comments from both reviewers are well addressed. I think it's ready to go. Yet, when I go through the revision in details, there are still some minor (very minor) writing flaws. Not critical, Not fatal. I've tried my best to do some proofreading (please see the attached file) but it may still need inputs from an native English speaker. I am looking forward to read this paper in DIVERSITY as soon as possible. 

Author Response

Reviewer1 Comments:

I am quite satisfied and appreciated to the present revision. It's flow is quite smooth and reads comfortable now. Congrats. Comments from both reviewers are well addressed. I think it's ready to go. Yet, when I go through the revision in details, there are still some minor (very minor) writing flaws. Not critical, Not fatal. I've tried my best to do some proofreading (please see the attached file) but it may still need inputs from an native English speaker. I am looking forward to read this paper in DIVERSITY as soon as possible.

The authors are very grateful for the advice, guidance and proofread of the article.

The authors revised the sentence according to the comments.

Do  these 'offshore development projects' indicate the 'One Thousand Turbine Project'? if it is, then I think using 'offshore wind-farm development' or 'One Thousand Turbine Project' will be better. When you talk about 'impact', you need to be specific.

Thanks for the suggestion. We revised to use ' offshore wind-farm development '

'have significantly increased' read awkward

  1. project can not 'increase' something, but the implementation of the project leads to the increasing something
  2. if you did not conduct a statistic analysis that shows vessel traffic do increase after something, you can not use the term 'significantly' here ('significant' always needs statistic evidence)

try: "leads to increasing ...."

Thanks for the suggestion. We revised to use ' leads to increasing '

endemic subspecies-----

(this is a 'subspecies' not yet an isolated 'species')

Thanks for reminding. We revised to use ' endemic subspecies '

Strikethrough respectively,

Removed it.

Strikethrough, respectively,

Removed it.

You need to be specific here. 'behavioral ecology' is a comprehensive term that includes feeding, mating, communication, socialization, ...etc. when you use 'behavioral ecology' here, it means everything.

try "...individual communication and feeding success of the population."

Thanks for the suggestion. We revise to use ' individual communication and feeding success of the population '

Strikethrough Further,

Removed it.

are proposed

We revise to use ' are proposed '

(Many Chinese graduates often use 'have/has verbed' to describe a fact that has been studied. in this kind of description, just 'verb' is ok because the description following 'Many studies show ...' is proved. In English, we use the simple present tense to describe the fact.

Indeed, the leading words 'Many studies show...." can be removed, just the description ('Vessel noise may result in xxxxx (references)', in this case), which makes your sentence succinct.

Thanks for the suggestion. We have rewritten this as

Vessel noise may result in behavioral disturbances of several species.

classified Critically Endangered in the IUCN Red List of Threatened Species

We have rewritten this as

The Taiwanese humpback dolphin, Sousa chinensis taiwanensis, is classified Critically Endangered in the IUCN Red List of Threatened Species

inhabits the

Thanks for the suggestion. We have rewritten this as

This endemic subspecies inhabits the coastal water of the Eastern Taiwan Strait, usually at a depth lower than 20 m

Strikethrough Since its first discovery in 2002, its The

Deleted it and added ‘The’

long-term viability

(in biological conservation, we don't discuss the 'growth' but the 'long-term viability' of a population).

Thanks for suggesting. We used ‘long-term viability’ to replace ‘further growth’

Strikethrough and .

Deleted ‘and’

Their social … population

(When you use 'survival' alone, in this context, you mean the survival individual that is independent with previous foraging and reproduction, i.e., 'therefore' should not be used here.) 'successful foraging and reproduction is the major factor influencing the 'population survival' (so you can use 'therefore' here.)

Thanks for suggesting. We have rewritten this as

Their social behavior is essential for successful foraging and reproduction and, therefore, population survival.

identify

We revised to use ‘identify’

Here, you talk about the 'Taiwanese humpback dolphin'. you should just cite the references related with Taiwanese humpback dolphin.

Removed ‘Tyack 2003’.

Strikethrough may,.

Deleted ‘may,’

indicate

We replaced ‘have shown’ as ‘indicate’

Orientation

We revised to use ‘orientation’

Strikethrough , respectively

Deleted ‘, respectively’

This present study showed …

We revised this as

This present study showed a significant drop in the whistling and clicking rates and significantly shorter whistles during and after vessel transit.

Strikethrough Evidence from previous studies suggests that

We revised this as

Changes in vocalization behavior may be due to acoustic interference, enhanced vigilance, reduced abundance, and stress.

describe

We replaced ‘have described’ as ‘describe’

, for example,

(you need 'for example' to keep the scope of your discussion on your context.

We revised this as

The Ganges river dolphin, for example, responds to vessel traffic and elevated noise levels by suppressing its acoustic activity

I am not going to change, proofread and edit here.

My comment is, however, I don't suggest you compose your discussion in this way (I know it's the training style for Chinese, including Taiwanese' graduates when they compose their discussion). In this way of description, the flow and scope of discussion often 'jumps' quickly. readers are hard to keep their focus and finally miss in texts.

In your next paper, I suggest you summarize those information you describe, such as

"In the presence of operating vessels, dolphins' acoustic behaviors change significantly, such as/including case 1 (references), case 2 (references)... and case n (references)." Then, following the discussion based on your own results. In this style, you make your discussion succince while readers can easily catch your points.

Thank you for your understanding and valuable suggestions.

It is very helpful for the authors to organize the discussion in the future.

Highlight had beed observed

We revised this as

Humpback dolphins significantly increased the whistle rate immediately when a boat passed through the area less than 1·5 km from the groups.

Strikethrough Taiwan Taiwanese…restricts…piling…of what

We revised this sentence as

The Taiwanese government restricts underwater noise to a sound exposure level (SEL) of no more than 160 dB (re 1μPa2?) at a distance of 750m from the piling.

is supposed…lower the risk of…

We revised this sentence as

To ensure this standard, noise mitigation measures such as bubble curtains are taken up, which is supposed to lower the risk of a temporary hearing threshold shift (TTS) or a permanent threshold shift (PTS) in humpback dolphins.

Strikethrough .Thus , which

We replaced ‘. Thus’ as ‘, which’

Strikethrough during activities (such as foraging and mating),

We deleted it.

rarely explored.

  1. 'unexplored' read awkward (Did you just do this exploration??)
  2. just tell reads this is a rarely explored topic (and readers will automatically imagine this require 'deeper study').

We replaced ‘unexplored areas that need deeper study’ as ‘rarely explored’

merit

We replaced ‘require’ as ‘merit’

development activities

We replaced ‘developmental projects’ as ‘development activities’

Reviewer 2 Report

The authors have addressed most of my comments, but there are still some problems with the literature cited.  Although, it has been improved, there are still inconsistencies in formatting (e.g., some journal names are upper case and some are lower case), and missing information (e.g., in some cases the journal name is missing).

Author Response

Reviewer2 Comments

The authors have addressed most of my comments, but there are still some problems with the literature cited.  Although, it has been improved, there are still inconsistencies in formatting (e.g., some journal names are upper case and some are lower case), and missing information (e.g., in some cases the journal name is missing).

The authors thank the reviewers for their suggestions and have reorganized the references according to the journal format
